# Synthesis and Characterization of the Donor-Acceptor Conjugated Polymer PBDB-T Implementing Group IV Element Germanium

**DOI:** 10.3390/polym15112429

**Published:** 2023-05-23

**Authors:** Wafaa H. Abousamra, Destinee Thomas, Dan Yang, Shahidul M. Islam, Cherese Winstead, Young-Gi Kim

**Affiliations:** Department of Chemistry, Delaware State University, Dover, DE 19901, USA

**Keywords:** electron donor and acceptor, conjugated polymers, fluorescent polymers, synthesis, group IV element, polymer solar cells

## Abstract

Here, we synthesized and characterized a novel two-dimensional (2D) conjugated electron donor–acceptor (D-A) copolymer (PBDB-T-Ge), wherein the substituent of triethyl germanium was added to the electron donor unit of the polymer. The Turbo–Grignard reaction was used to implement the group IV element into the polymer, resulting in a yield of 86%. This corresponding polymer, PBDB-T-Ge, exhibited a down-shift in the highest occupied molecular orbital (HOMO) level to −5.45 eV while the lowest unoccupied molecular orbital (LUMO) level was −3.64 eV. The peaks in UV-Vis absorption and the PL emission of PBDB-T-Ge were observed at 484 nm and 615 nm, respectively.

## 1. Introduction

Polymer solar cells (PSCs), comprising bulk-heterojunction (BHJ) photoactive layers of electron donors (D) and electron acceptors (A), have attracted enormous attention due to their lightweight properties, solution-processability, low-temperature printing fabrication and mechanical flexibility [1,2,3,4,5,6,7,8]. Several types of electron-acceptor materials have been developed for use in BHJ PSCs, including fullerene and non-fullerene derivatives, which cover allotropes, polymers, and small organic molecules (SMAs) [7,8,9,10,11,12,13,14,15,16,17,18,19,20,21,22,23,24]. Fullerene derivatives were reported to exhibit attractive properties that made them promising electron acceptors for PSCs in the past. These characteristics of the fullerene derivatives include affordable three-dimensional electron mobility, forming appropriate phase-separation in the photoactive layer [24,25,26]. However, despite the outstanding performance of fullerene-based PSCs, fullerene derivatives still exhibit many intrinsic drawbacks such as high production costs with tedious purification steps, weak absorption in the visible spectral region, and the limited tunability of energy level [24,25,26]. 

Other electron acceptors, such as SMAs, have shown advantages in simple synthesis methods with various structural variations to acquire favorable absorption ranges and modulate frontier energy levels [24,25,26,27,28,29]. Due to the advantages of SMAs, the power-conversion efficiencies (PCEs) of the SMA-based PSCs were reported to have continuously improved by ca. 17% [6,30,31,32]. Considerable efforts have been made to construct high-performance non-fullerene (NF) PSCs that match the electronic band structure of electron donors and electron acceptors to absorb photons in the visible and near-infra red regions. The structural modification of electron donors and acceptors can improve short-circuit currents (J_sc_), and the alignment of the energy levels between electron donors and acceptors was reported to control open-circuit voltage (V_oc_). The crystallinity and morphology of electron donor–acceptor pairs have also been reported to impact the fill factor (FF) [7,32,33]. 

PSCs use the variable bandgap conjugated polymer to construct the BHJ photoactive layer, modulating electron-donating (D) and electron-accepting (A) characteristics of the conjugated D-A polymers. The molecular engineering of D-A moieties in the backbone of the polymers is considered to be one of the most successful strategies to tune the energy levels and adjust optical properties [4,33]. Recently, many types of D-A copolymers have been developed for the photoactive layer of PSCs and reported to show a PCE of 16% or higher [1,30].

Recently, Bin H. et al. adopted a new strategy to create copolymers, tuning the highest occupied molecular orbital (HOMO) by implementing the element Si, one of the group IV elements, in the backbone of the BDTT-FBTA (J71) copolymer [34]. Replacing the alkyl group with the trialkyl silyl substituent decreased the HOMO to −5.4 eV, enhancing the absorption, which was reflected by the achievement of a PCE of 11.4%. Previously, analogous alkyl sidegroup-based PSC exhibited a PCE of 9.5%. To examine this idea, Chen S. et al. synthesized several D-A copolymers, coupling BDTT-Si with different acceptor moieties. The PSCs were reported to show a PCE of 12.1% [33].

Motivated by previous studies that implemented the group IV elements in the backbone of the D-A-conjugated copolymers, we investigated another group IV element, germanium. It is expected to have stronger *π-π* stacking due to the longer C–Ge bond length compared with the C–C and C-Si bonds. It is known to assist in intermolecular charge transport and suppress the charge carrier recombination, which eventually improves the photovoltaic performance. Here, we have synthesized and characterized a new copolymer, PBDB-T-Ge, using the substituent of trialkyl germane on the BDTT (BDTT-Ge) moiety, which is the donor unit of the copolymer and is coupled with the acceptor moiety BDD. Figure 1 illustrates the molecular structures of PBDB-T-Ge and precedent copolymers, PBDB-T-C and PBDB-T-Si, where the sidegroups of the electron donor of the polymers are trialkyl germane, ethyl hexyl and tripropyl silyl substituent, respectively. In this paper, we focus on the synthesis and characterization of the PBDB-T-Ge to compare the electronic band structure. 

## 2. Materials and Experimental Methods

### 2.1. Materials

Dichloromethane (DCM, 99.9%), isopropyl magnesium chloride lithium chloride complex (iPrMgCl·LiCl, 1.3 M in THF), trimethyl tin chloride (1 M in hexane, 25 wt.%), acetone and deuterated chloroform (*^d^*CDCl_3_) were purchased from Acros Organics (Branchburg, NJ, USA). Hexane, methanol, toluene, hydrochloric acid (HCl), chloroform, acetonitrile and ethyl acetate were purchased from Fisher Scientific (Branchburg, NJ, USA). Ferrocene, tetraethylammonium tetrafluoroborate (Et_4_NBF_4_) and diethyl ether were obtained from Alfa Aesar (Ward Hill, MA, USA). Anhydrous tetrahydrofuran (THF), n-Butyllithium solution (n-BuLi, 2.5 M in hexane), Benzo [1,2-*b*:4,5-*b*′]dithiophene-4,8-dione, tin(ll) chloride dihydrate, anhydrous sodium sulfate (>99%), indium tin oxide (ITO) glass substrate, 1,3-Bis(5-bromo-2-thienyl)-5,7-bis(2-ethylhexyl)-4*H*,8*H*-benzo [1,2-*c*:4,5-*c*′]dithiophene-4,8-dione (BDD, 97%), tetrakis(triphenylphosphine) palladium (Pd(PPh_3_)_4_) and Y6 were purchased from Sigma-Aldrich (Burlington, NJ, USA). Triethyl germanium chloride and 2-iodothiophene were obtained from TCI (Boston, MA, USA). Ethanol was obtained from EMD (Burlington, NJ, USA). Isopropyl alcohol was purchased from VWR (Missouri City, TX, USA). All materials were used as received without further purification.

### 2.2. Synthesis of PBDB-T-Ge 

#### 2.2.1. Synthesis of Triethyl (Thiophen-2-yl) Germane (**2**)

Triethyl(thiophen-2-yl) germane was synthesized via a method that was adopted from Sherborne G. et al. [35]. Under the protection of argon, THF (141 mL) was placed in a 500 mL, two-neck, round-bottom flask. Triethyl germanium chloride (5.5 g, 4.7 mL, 28.16 mmol) and 2-iodothiophene (5.38 g, 2.83 mL, 25.6 mmol) (**1**) were added to the round-bottom flask under stirring condition. Then, iPrMgCl·LiCl (23.6 mL, 30.73 mmol, 1.3 M in THF) was added dropwise to the reaction mixture, which was left under stirring conditions for 3 h at room temperature. The mixture was quenched with cold DI water (30 mL), which was then extracted by DCM (25 mL) three times. The organic layer was collected and dried over anhydrous sodium sulfate (Na_2_SO_4_). The volatile solvent was removed under vacuum with mild temperature; then, the residue was purified by column chromatography using hexane: ethyl acetate (5:1 *v*/*v*) as an eluent with a retention factor (RF) of 0.0789 to yield the triethyl(thiophen-2-yl) germane as a colorless oil. Yield = 5.4 g (86%). ^1^H NMR (400 MHz, CDCl_3_), δ (ppm): 7.62–7.59(d, 1H), 7.25–7.22(m, 1H), 7.20–7.19(d, 1H), 1–1.3(m, 15H). 

#### 2.2.2. Synthesis of 4,8-Bis(5-triethyl germanium)thiophene-2-yl)benzo[1,2-*b*:4,5-*b*′]dithiophene (**3**)

Under the protection of argon, a solution of triethyl(thiophen-2-yl) germane (2.5 g, 10.29 mmol, compound **2**) in THF (45 mL) was placed in a 100 mL, two-neck, round-bottom flask. The solution was cooled to 0 °C. n-BuLi (4.12 mL, 10.29 mmol, 2.5 M in hexane) was added dropwise into the RBF and the mixture was kept at 0 °C for 40 min before the mixture was refluxed at 65 °C under stirring for 2 h. After we cooled the mixture down, 4,8-dehydrobenzo[l,2-*b*:4,5-*b*′] dithiophene-4,8-dione (0.75 g, 3.43 mmol) was added in one portion, and the mixture was refluxed with stirring for another 2 h. Tin(ll) chloride dihydrate (6.19 g, 27.44 mmol) in 10% HCl (15 mL) was added after the mixture was cooled down to room temperature and stirred overnight. The reaction was quenched using cold DI water (40 mL), and then the crude product was extracted using diethyl ether (25 mL) three times. The organic layer was collected and dried over anhydrous sodium sulfate (Na_2_SO_4_). The volatile solvent was removed under vacuum at a mild temperature and the crude product was purified using a column chromatography with hexane: ethyl acetate (5:1 *v*/*v*) as eluents to yield 4,8-bis-(5-(triethyl germanium)thiophene-2-yl)benzo[1,2-*b*:4,5-*b*′]dithiophene as bright yellow gel. Yield = 0.8 g (36.2%). ^1^H NMR (400 MHz, CDCl_3_), δ (ppm): 7.67–7.65(d, 2H), 7.6–7.58(d, 2H), 7.47–7.45(d, 2H), 7.3–7.28(d, 2H), 1.00–1.4(m, 30H). 

#### 2.2.3. Synthesis of [4,8-Bis[5-triethyl germanium) thiophene-2-yl]-2-trimethylstannyl thieno[2,3-f][1]benzothiol-6-yl]-trimethylstannane (**4**)

Under the protection of argon, a solution of 4,8-bis-(5-(triethyl germanium) thiophene-2-yl)benzo[1,2-*b*:4,5-*b*′]dithiophene (0.4 g, 0.596 mmol) in THF (6 mL) was placed in a 25 mL round-bottom flask. The solution was cooled to −78 °C using dry ice with acetone, and then n-BuLi (0.596 mL, 1.49 mmol, 2.5 M in hexane) was added dropwise. After the addition, the mixture was kept at −78 °C for 1 h; trimethyl tin chloride (1.78 mL, 1.78 mmol, 1 M in hexane) was added dropwise. The resulting mixture was stirred overnight at room temperature. The mixture was then poured into cold DI water (6 mL) and extracted with diethyl ether (8 mL) three times; after drying over anhydrous sodium sulfate (Na_2_SO_4_), the volatile solvent was removed under vacuum at a mild temperature. Recrystallization was used for purification using ethanol as a solvent, and then dried for two days in a vacuum at a mild temperature to yield a yellowish white crystal of [4,8-Bis[5-triethyl germanium)thiophene-2-yl]-2-trimethylstannylthieno[2,3-f][1]benzothiol-6-yl]-trimethylstannane. Yield = 0.092 g (15.6%). ^1^H NMR (400 MHz, CDCl_3_), δ (ppm): 7.49(s, 2H), 7.48–7.46(d, 2H), 7.2– 7.3(d, 2H), 1.2–0.9(m, 30H), 0.6–0.5(m, 18H). 

#### 2.2.4. Synthesis of PBDB-T-Ge (**6**)

In a 25 mL, double-neck, round-bottom flask, [4,8-Bis[5-triethyl germanium)thiophene-2-yl]-2-trimethylstannylthieno[2,3-f][1]benzothiol-6-yl]-trimethylstannane (50 mg, 0.08 mmol) and 1,3-Bis(5-bromo-2-thienyl)-5,7-bis(2-ethylhexyl)-4*H*,8*H*-benzo[1,2-*c*:4,5-*c*′]dithiophene-4,8-dione (58 mg, 0.08 mmol) were dissolved in dry toluene (5 mL). The reaction container was purged with argon for 20 min, and then tetrakis(triphenylphosphine)palladium (Pd(PPh_3_)_4_) (12 mg, 0.01 mmol) was added. After another flushing with argon for 20 min, the reactants were heated up to reflux them at 120 °C for 24 h. The solution was cooled down to room temperature and poured into cold methanol (MeOH) (50 mL), and then washed three times using cold methanol. The product was dried under vacuum at a mild temperature for 24 h to yield a dark purple solid of PBDB-T-Ge. Yield 0.09 g. Figure 1 illustrates the synthetic route of PBDB-T-Ge. Mn and Mw of PBDB-T-Ge were measured to be 2.8 KDa and 3.9 KDa using GPC (CHCl_3_), respectively. The polydispersity (PDI) was observed to be 1.41. ^1^H NMR (400 MHz, CDCl_3_), δ (ppm): 7.6–7.75(d, 2H), 7.45–7.35(d, 2H), 7.2–7.3(d, 2H), 7.15–7.25(d, 2H), 6.95–7.05(t, 2H), 1.4–0.95(m, 30H), 0.9–0.6(m, 34H).

### 2.3. Characterization

The structure of PBDB-T-Ge was confirmed using an NMR spectrometer (Varian Unity Inova, 400 MHz). The spectra were collected using *d*-chloroform as solvent and the internal chemical shift reference standard was selected based on the residual proton resonance of the solvent, which were acquired, processed, and analyzed using Bruker Topspin (4.1.4) software. The molecular weight of polymers was determined by gel permeation chromatography (GPC) relative to polystyrene standards with chloroform as the eluent at a concentration of 1.0 mg/ mL for at least 12 h. The optical properties were tested using a UV-Vis spectrometer (Shimadzu 3600 Plus). This characterization was performed using chloroform solution in a quartz cuvette using polymer film on a glass substrate. The polymer film samples were prepared using drop-casting on a glass substrate. The photoluminescence study was performed using Spectro-fluorophotometer (Shimadzu RF-5301PC model) using a chloroform solution. Cyclic voltammetry study was performed using a potentiostat (CH instrument electrochemical workstation, model CHI 660C) at room temperature to determine the redox reaction. The surface morphology characterizations on PBDB-T-Ge:Y6 (1:1.2, *w*/*w*) composite film were performed using atomic force microscopy (AFM, Bruker Inova) using a taping mode for imaging. Further surface morphology was performed using scanning electron microscopy (SEM, FEI. Quanta FEG 250) with secondary electron for the imaging.

## 3. Results and Discussion

### 3.1. Synthesis of PBDB-T-Ge

Our work implemented germanium on the side branch of the PBDB-T-Ge. Previous reports utilized *n*-butyl lithium to activate the ethylhexyl branch groups of PBDB-T-C and trialkyl silyl groups of PBDB-T-Si, as seen in Figure 1. Although this method is successful for alkylation on the backbone of polymers, it requires an extremely low temperature of −78 °C and an extended reaction time with overnight stirring [34,36]. In this report, we adopted a new method to synthesize PBDB-T-Ge, introducing a sidegroup of trialkyl germanium using a Turbo–Grignard reagent (iPrMgCl·LiCl), which allowed for alkylation to occur at room temperature and the reaction time was cut down to three hours [35]. Additionally, using the Turbo–Grignard reaction led to a higher yield of 86%, while the conventional alkylation methods (BuLi method) produced a lower yield of 45.3%. The Stille C-C polycondensation reaction was used to polymerize PBDB-T-Ge using comonomers of BDD and BDTT-Ge, which are an electron-acceptor and electron-donor, respectively [37].

^1^H NMR spectra in Figure 2 show that triethyl(thiophen-2-yl) germane (**2**) is synthesized, where the three peaks between 7.1 and 7.7 ppm are assigned to the proton of the single-branched thiophene and the multiple peaks from 0.8 to 1.4 ppm are assigned to the alkyl groups. The ^1^H NMR spectra show that the implementation of triethyl germanium on the thiophene was successful, simplifying and enhancing product yields.

### 3.2. UV-Vis Absorption of PBDB-T-Ge

Figure 3 shows the UV-visible absorption characteristics of comonomers and polymers, which are BDTT-Ge (M1), BDD (M2), PBDB-T-Ge, respectively. In a chloroform solution, the comonomers BDTT-Ge and BDD showed a defined absorption profile with single peaks at 385 nm and 445 nm, respectively, while PBDB-T-Ge exhibited two peaks-absorption profiles at 359 and 484 nm. The red shift in the maximum absorption peaks of the co-monomers confirms successful polymerization, showing extended conjugation.

The UV-Vis absorption spectra of the PBDB-T-Ge solution in chloroform and the polymer film on a glass substrate were compared, as shown in Figure 4, where the absorption of the polymer solution appears to be between 410 and 630 nm with an absorption onset (λ_onset_) at 620 nm, and the maximum absorption (λ_max_) occurred at 477 nm. In comparison, the absorption peak of reference polymer (PBDB-T-C, Figure 1) with alkyl substituent [38] was reported to be between 500 and 650 nm, showing a maximum absorption peak at 618 nm. The absorption peak of PBDB-T-Ge film was found to appear between 420 and 680 nm, with an onset (λ_onset_) of 666 nm and maximum absorption (λ_max_) at 507 nm, while another reference polymer film (PBDB-T-Si, in Figure 1) [33] showed an absorption peak between 500 and 670 nm. The UV-Vis absorption peak of the commercial electron acceptor, Y6, in thin film appeared between 580 and 920 nm, with λ_max_ of 838 and λ_onset_ of 920 nm. PBDB-T-Ge and Y6 seems to exhibit a complementary absorption, which is beneficial for light harvesting, making them a good candidate for use in a blend film in non-fullerene PSC. The optical properties of PBDB-T-Ge are summarized in Table 1, along with the emission profile. The onset absorption and maximum UV-Vis absorption in solution and the film for PBDB-T-Ge and Y6 are shown in Figure 4. It is noticeable that there is a red shift of 30 nm from solution to film, indicating that there are stronger *π-π* intermolecular interactions in the film with a better stacking structure.

### 3.3. Photoluminescence of PBDB-T-Ge

PBDB-T-Ge solution showed a red–orange–fluorescent light at an excitation of 470 nm, as seen in Figure 5. The emission peak appeared in the range between 550 and 740 nm, with the maximum emission peak shown at 615 nm. The intensity of the emission peak in the polymer was observed to be suppressed by the addition of the electron-acceptor PC_60_BM solution in chloroform, as shown in Figure 5. The fluorescence-quenching explains that the photo-induced charge transfer occurred between the electron-donating PBDB-T-Ge and the electron-accepting PC_60_BM.

### 3.4. Electrochemical Characteristics of PBDB-T-Ge

The HOMO and LUMO of PBDB-T-Ge were measured to evaluate the effect of the substitution of the triethyl germane sidegroup on the electronic energy levels using cyclic voltammetry. The electrochemical study was performed using conventional three electrodes configuration consisting of an Ag/AgCl electrode as a reference electrode, the glassy carbon electrode as the working electrode, and the platinum wire as the counter-electrode. We used tetraethylammonium tetrafluoroborate (Et_4_NBF_4_, 0.1 M) in acetonitrile as the supporting electrolyte and the ferrocene/ferrocenium as the internal reference. Cyclic voltammogram was obtained at a scan rate of 10 mV/s with a 1 mV sample interval.

Solutions of PBDB-T-Ge polymer and Y6 were prepared by dissolving in chloroform and the films were deposited onto the glassy carbon button electrode via a drop-casting procedure. It was then air-dried. HOMO and LUMO energy levels were estimated from onset oxidation and reduction potentials (E_ox_^onset^ and E_red_^onset^), respectively, according to the Equations (1) and (2) [36].
E (HOMO) = −e [E_ox_^onset^ + 4.8 − E_1/2_.Fc/Fc^+^](1)
E (LUMO) = −e [E_red_^onset^ + 4.8 − E_1/2_.Fc/Fc^+^](2)
where E_1/2_.Fc/Fc^+^ is the half-potential of ferrocene/ferrocenium coupled to the electrochemical measurement system, and the energy level of Fc/Fc^+^ was taken as 4.8 eV below vacuum. Ferrocene/ferrocenium half-potential was measured to be 0.44 V ferrocene/ferrocenium, and Equations (3) and (4) are as follows:E (HOMO) = −e [E_ox_^onset^ + 4.36](3)
E (LUMO) = −e [E_red_^onset^ + 4.36](4)

Figure 6a shows the cyclic voltammogram of PBDB-T-Ge film, from which the onset oxidation potential (E_ox_^onset^) and onset reduction potential (E_red_^onset^) of PBDB-T-Ge were measured to be 1.09 and −0.72 V vs. Ag/AgCl, respectively. The HOMO energy level and LUMO energy level of PBDB-T-Ge were calculated to be −5.45 and −3.64 eV, respectively, as seen in Figure 6b. Under the same experimental conditions, the HOMO and LUMO energy levels of Y6 were found to be −5.65 eV and −3.86 eV, respectively. The LUMO energy offsets between PBDB-T-Ge as donor and Y6 as acceptor were larger than the exciton-binding energy, which ensures efficient photo-induced electron transfer.

We noticed that the HOMO of PBDB-T-Ge was deepest among the three polymers and showed a smaller gap of 0.22 eV and 0.31 eV compared to PBDB-T-C and PBDB-T-Si, respectively. The small band gap is well-known to improve J_sc_, which is expected to enhance the photovoltaic performance. The data were collected from PBDB-T-C in 0.1 M tetrabutylammonium hexafluorophosphate (Bu_4_PF_6_) acetonitrile solution at a scan rate of 50 mV/s, while the data of PBDB-T-Si were collected using Ultraviolet Photoelectron Spectroscopy (UPS). The detailed electrochemical data for PBDB-T-Ge and its counterparts, Y6, are listed in Table 2, which indicates that the triethyl germane substitution with σ* (Ge) – π* (C) bond interaction is an effective way to lower the HOMO energy level of 2D-conjugated polymers. Lowering the HOMO energy level is beneficial to improve V_oc_, which is expected to enhance the PCE of PBDB-T-Ge-based PSC. It should be noted that the electrochemical bandgap (E_g_**^cv^**) of PBDB-T-Ge is 1.81 eV, which is similar to its optical bandgap of 1.86 eV.

### 3.5. Morphology Study

Molecular-disorder-induced nanostructure variations in the BHJ are critical in optimizing photo-induced charge separation and transportation in OSCs. The surface morphology study and characterizations of PBDB-T-Ge:Y6 (1:1.2, *w*/*w*) blend film was performed using atomic force microscopy (AFM) and electron-scanning microscopy (SEM), as depicted in Figure 7. The biphasic domain of the PBDB-T-Ge:Y6 composite film can clearly be seen in the morphology study.

### 3.6. Structural Optimization and Density Functional Theory (DFT) Calculations

Density functional theory (DFT) calculations were conducted to provide further insights into the fundamental aspects of the molecular architecture of germanium-based comonomers. All calculations were conducted with the Gaussian16 software package. Optimized geometries of all compounds were obtained at the B3LYP level of theory with the 6-31G(d) basis set for all atoms. Previous work [39,40,41] showed that the geometries, and energetics of similar systems obtained using the B3LYP level of theory and 6-31G(d) basis set showed better agreement with experiments. Frequencies were calculated to ensure the absence of imaginary frequencies in the lowest-energy state. The HOMO and LUMO Hartree energies obtained from the calculation were converted to eV. The electrical transport properties of a molecule depend on the energy gap of HOMO and LUMO orbitals. The calculated frontier molecular orbital structures of the complexes are presented in Figure 8.

We observed that the electron density distributions at the HOMO are highly localized on the electron-rich moiety of BDTT-Ge (M1) in the ground state while, in the excited state, it is distributed at the LUMO and highly localized in the electron-deficiency moiety BDD (M2). The transition of the electron density distributions at the HOMO and LUMO were observed from all cases of -C, -Si and -Ge.

DFT calculations of Ge-based comonomer (triethyl-Ge) estimated the HOMO of −5.02 eV and LUMO of −2.36 eV with the band gap of 2.66 eV, while the band gaps for the counter-partner C-based comonomer (triethyl-C) and Si-based comonomer (triethyl-Si) were calculated to be 2.655 and 2.68 eV, respectively. The calculated electronic energy values of the HOMO, LUMO and band gap of the three co-monomers are summarized in Table 3. Clearly, the transition of charge density observed at LUMO and HOMO shows that the conjugated D-A comonomer can be a promising candidate for PSC, allowing for an efficient photo-induced intramolecular charge transfer (ICT) effect of the molecules.

## 4. Conclusions

A new 2D-conjugated donor–acceptor (D-A) copolymer (PBDB-T-Ge) was synthesized, where two reaction methods were compared to implement triethyl Ge: the conventional n-BuLi basis and Turbo–Grignard reactions. The Turbo–Grignard reaction was observed to be efficient in incorporating the group IV element to the side chain with a yield of 86%. The PBDB-T-Ge exhibited a LUMO of −3.64 eV, which was closer to that of Y6, an electron-acceptor. The gap between the LUMO values was calculated to be ca. 0.22 eV, which is sufficient to expect efficient photo-induced charge injection between the electron-donor and -acceptor in BHJ PSCs. PBDB-T-Ge and Y6 showed a complementary absorption, indicating a wide range of light-harvesting. DFT calculations on the comonomer verified that there were no significant changes in the band gap when the Ge sidegroup of the polymer replaces carbon- and silicon-base sidegroups.

## Data Availability

The data presented in this study are available on request from the corresponding author.

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
