# Peer review of "Synthesis and Characterization of the Donor-Acceptor Conjugated Polymer PBDB-T Implementing Group IV Element Germanium"

_polymers, 2023, doi:10.3390/polym15112429_

Round 1

Reviewer 1 Report

In this study, the authors synthesized and characterized a novel two-dimensional electron donor-acceptor copolymer named PBDB-T-Ge. They incorporated a substituent of triethyl germanium onto the polymer's electron donor unit using the Turbo-Grignard reaction, with a yield of 86%. To thoroughly investigate the properties of the copolymer, the authors employed various characterization methods, including NMR, UV-vis, Photoluminescence, CV, AFM, and SEM. Additionally, the authors conducted a DFT study to further analyse the copolymer's characteristics. The manuscript presented by the authors requires significant revision to improve its scientific quality. Below are specific comments:

• The authors should either test their materials in PSC or explain how their materials enhance PSC performance, as it is not clear from the results and discussions.

• Since the authors have already performed DFT calculations, they should theoretically calculate the Jsc and Voc of PSC with their materials to at least get an idea of whether the materials are following the expected trend.

• There are contradictions between the onset oxidation and reduction values for PBDB-T-Ge in the text and Fig 6(a). Additionally, the graph for PBDB-T-Ge in Fig 6(a) is too small to be clearly visible.

• The relevant references should be cited in the caption of Fig 6(b).

• The SEM image in Fig 7 is not clear and should be replaced with a clearer one.

• Some methods appear in the results and discussion section and should be corrected (e.g., line no. 232-239 and line no. 296-306).

• The same acronyms (e.g., PSC, D and A) are introduced multiple times in the same section and should be avoided.

Author Response

  • The authors should either test their materials in PSC or explain how their materials enhance PSC performance, as it is not clear from the results and discussions. Since the authors have already performed DFT calculations, they should theoretically calculate the Jsc and Voc of PSC with their materials to at least get an idea of whether the materials are following the expected trend.
    • This manuscript is focused on the synthesis and characterization of PBDB-T-Ge.
    • Authors hope to prepare separate manuscript of PSC device (including DFT calculation for PSC performance) using this polymer in the future.  
    • The scope of this paper is explained in line 62-72 along with the expectation.
  • There are contradictions between the onset oxidation and reduction values for PBDB-T-Ge in the text and Fig 6(a). Additionally, the graph for PBDB-T-Ge in Fig 6(a) is too small to be clearly visible.
    • The negative signs in the lines 272 and 273 have been fixed. The size of Fig 6 (a) has been increased. The Fig 6 (b) has been fixed and enlarged. The number in Table 2 is fixed.
  • The relevant references should be cited in the caption of Fig 6(b).
    • Reference has been added to Fig 6 and Table 2.
  • The SEM image in Fig 7 is not clear and should be replaced with a clearer one.
    • The picture in the manuscript is the best quality picture we have.
  • Some methods appear in the results and discussion section and should be corrected (e.g., line no. 232-239 and line no. 296-306)
    • Please specify the parts to be corrected. We will follow up accordingly.
  • The same acronyms (e.g., PSC, D and A) are introduced multiple times in the same section and should be avoided.
    • PSCs has been fixed at line 44. D and A at line 20 were removed to avoid confusion.

Reviewer 2 Report

In the manuscript the authors reported synthesis and general characterization of PBDB-T-Ge having the group IV element germanium implemented onto the donor unit of the polymer as triethyl germanium substituent.  Then they compared optoelectrical properties of PBDB-T-Ge with two other analogues having carbon (PBDB-T-C) and silicon (PBDB-T-Si) substitutions. They also looked at blends of PBDB-T-Ge and Y6 to show that they can be good combination for OSCs. The main strong point of this research is to use the Turbo-Grignard reaction to produce triethyl (thiophen-2-yl) germane. Although the design and synthesis of PBDB-T-Ge is original and thus if of interest for the community, I found characterization of the polymer, monomer and its precursor molecules is not adequate. Therefore, I would like the authors to provide following points before publication is recommended.

1. For the synthesis of PBDB-T-Ge, including the monomer its precursors and polymer itself, only proton NMR is provided. Some more general characterization such as 13C-NMR, and elemental analysis or mass spectrum should be added. Also, for PBDB-T-Ge its yield, molecular weights and dispersity data are missing. For the molecules which were purified with column chromatography, please also provide their Rf values.

2. I feel like the title of the manuscript can be improved so that it can better represent the research. I would suggest this or something similar: ‘Synthesis and Characterization of the Donor-Acceptor Conjugated Polymer PBDB-T Implementing Group IV Element Germanium’.

3. L45, the conjugated D-A polymers;

4. L92, added dropwise;

5. In Figure 4, I suggest to add spectra of the PBDB-T-Ge: Y6 blend film as well.

6. In Figure 6 caption, please provide the references from which the energy values of PBDB-T-C and PBDB-T-Si were taken.

7. In Table 2, please provide the methods that were used to collect HOMO-LUMO energy levels for the other two polymers and Y6 as well.

8. For Figure 7(a) please add units.

9. In Figure 8 caption and the section of DFT calculation, it was said that PBDB-T-Ge, PBDB-T-C, and PBDB-T-Si were calculated. But what is actually calculated was the only one repeat unit. Therefore, please remove ‘P’ in all cases.

10. Table 3 caption can be improved: Electronic Characteristics of PBDB-T with Different Side Groups from DFT Calculations.

Thank you!

Author Response

  • For the synthesis of PBDB-T-Ge, including the monomer its precursors and polymer itself, only proton NMR is provided. Some more general characterization such as 13C-NMR, and elemental analysis or mass spectrum should be added. Also, for PBDB-T-Ge its yield, molecular weights and dispersity data are missing. For the molecules which were purified with column chromatography, please also provide their Rf values.
    • RF of Triethyl(thiophen-2-yl) germane has been added at line 102. molecular weights and dispersity data have been added to line 152-153. 13C NMR and elemental analysis are not available.
  • I feel like the title of the manuscript can be improved so that it can better represent the research. I would suggest this or something similar: ‘Synthesis and Characterization of the Donor-Acceptor Conjugated Polymer PBDB-T Implementing Group IV Element Germanium’.
    • The title has been changed as suggested.
  • L45, the conjugated D-A polymers.
    • It is changed.
  • L92, added dropwise.
    • It is fixed.
  • In Figure 4, I suggest to add spectra of the PBDB-T-Ge: Y6 blend film as well.
    • The absorption spectra of PBDB-T-Ge:Y6 blend film are not available.
  • In Figure 6 caption, please
    • References have been added.
  • In Table 2, please provide the methods that were used to collect HOMO-LUMO energy levels for the other two polymers and Y6 as well.
    • The methods for the other two reference polymers have been added at line 290-293. The data of Y6 was collected in our lab using the same methods for PBDB-T-Ge.
  • For Figure 7(a) please add units.
    • Added the unit.
  • In Figure 8 caption and the section of DFT calculation, it was said that PBDB-T-Ge, PBDB-T-C, and PBDB-T-Si were calculated. But what is actually calculated was the only one repeat unit. Therefore, please remove ‘P’ in all cases.
    • It is fixed.
  • Table 3 caption can be improved: Electronic Characteristics of PBDB-T with Different Side Groups from DFT Calculations.
    • The table caption has been changed.

Reviewer 3 Report

The authors report a novel two-dimensional conjugated electron donor-acceptor copolymer (PBDB-T-Ge). The inclusion of metal atom into polymer donors is an interesting subject, because the metal atom may stabilize the material photostability, suppress the internal conversion, etc. However, the authors only give few measurements for PBDB-T-Ge, which are AFM, CV, and PL. At least, an organic solar cell based on PBDB-T-Ge should be presented and compared to those based on PBDB-T-C and PBDB-T-Si.

The major revisions are needed.

Author Response

The followings are revised to improve the manuscript:

  • The scope of this paper is clearly explained in line 62-72 along with the expectation.
  • The negative signs in the lines 272 and 273 have been fixed.
  • The size of Fig 6 (a) has been increased.
  • The Fig 6 (b) has been fixed and enlarged.
  • The number in Table 2 is fixed.
  • Reference has been added to Fig 6.
  • RF of Triethyl(thiophen-2-yl) germane has been added at line 102.
  • Molecular weights and dispersity data have been added to line 152-153.
  • The title has been changed as follows: ‘Synthesis and Characterization of the Donor-Acceptor Conjugated Polymer PBDB-T Implementing Group IV Element Germanium’
  • References have been added to Figure 6.
  • The methods for the other two reference polymers have been added at line 290-293. The data of Y6 was collected in our lab using the same methods for PBDB-T-Ge.
  • In Figure 8, the figure captions were changed including comonomers.

Round 2

Reviewer 2 Report

Dear Athours,

Thanks for making reversion.I have no further comments. However, 13C NMR data are still missing. Hope you could still provide that.

Thank you 

Author Response

13C NMR data is not available.

Reviewer 3 Report

The manuscript can be accepted for the publication. In the abstract, the LUMO level should be expressed in minus. Please edit the entire paper.

Author Response

The minus sign is added.